# Factors Impacting the Sustainable Development of Professional Learning Communities in Interdisciplinary Subjects in Chinese K-12 Schools: A Case Study

**Yiyun Hu [1], Xiaoli Jing [1,*] and Yaqing Yang [2]**

1   Institute of International and Comparative Education, Beijing Normal University, Beijing 100875, China
2   Nanjing Yangzi No. 4 Primary School, Nanjing 210044, China
*   Correspondence: jxltxszh@126.com

**Abstract:** Professional learning communities are recognized as one of the most effective approaches for promoting the professional development of teachers. In the current complex and rapidly changing era, to facilitate the implementation of interdisciplinary curricula, Chinese schools have made tremendous efforts to enhance teacher professional development, particularly by establishing professional learning communities. Aiming to understand the operation of professional learning communities in interdisciplinary subjects in Chinese K-12 schools and to examine factors impacting the sustainable development of these professional learning communities, we conducted a case study on professional learning communities in the interdisciplinary subject of Education for International Understanding in Chengdu Horsens Primary School. As part of this study, we interviewed the principal, course director, seed teachers and teachers participating in the selected case. The research results demonstrated that the major factors impacting the sustainable development of professional learning communities in interdisciplinary subjects in Chinese K-12 schools include school structures and policies, school leadership, teachers' professionalism and learning capacity and their sense of community. In addition, compared to traditional subject-based professional learning communities in China, professional learning communities in interdisciplinary subjects highlight a sense of community, which presents three distinctive features: a conflict-inclusive atmosphere, the coexistence of individual and shared visions and an emotional bonding identity. These three features also have a considerable impact on the sustainable development of professional learning communities in interdisciplinary subjects in Chinese K-12 schools.

**Keywords:** professional learning communities in interdisciplinary subjects; teachers' professional development; teacher learning; sustainable development; China



## 1. Introduction

Teachers are one of the most influential and powerful forces for promoting educational equity and high-quality educational development. They are also one of the key factors in advancing sustainable development. In Sustainable Development Goal 4, which aims to "ensure inclusive and equitable quality education and promote lifelong learning opportunities for all", the United Nations states that one of its targets is to "increase the supply of qualified teachers in developing countries" [1]. Likewise, the United Nations Educational, Scientific and Cultural Organization has made the supply of well-trained, supported and qualified teachers one of its top priorities [2]. Nowadays, learning has been widely recognized as an indispensable approach for improving teacher quality and promoting teacher professional development. According to Andreas Schleicher [3], Director for the Directorate of Education and Skills at the Organisation for Economic Co-operation and Development, in the globalized era and the knowledge-based economy, approaches for promoting teacher learning have undergone radical changes with the conceptual shift from individual learning to professional community learning [4]. In the last three decades, professional learning

communities (hereafter PLCs) have been developed as one of the most effective approaches for promoting teacher professional development. By improving peer-to-peer communication, facilitating collaborative learning and building shared responsibilities, PLCs have a systematic and positive impact on teacher professional development [5].

*1.1. Literature Review*

Our literature review revealed that educational researchers have examined several issues related to PLCs, which can be roughly classified into two categories. The first category is the definition, attributes and functions of PLCs. PLCs are defined differently due to the varying social, cultural and educational backgrounds in different countries [6]. The most prevalent definition of PLCs is that proposed by Hord [5], who describes the PLC as a professional community of learners in which the teachers in a school, along with its administrators, continuously seek and share learning, as well as acting on their learning. The goal of their actions is to enhance their effectiveness as professionals for the benefit of students. Regarding the attributes of PLCs, Hord and Sommers [7] identify PLCs as having five characteristics: shared values, visions and goals, collective learning and application, shared individual practice, shared and supportive leadership and supportive conditions. Studies on the functions of PLCs highlight that well-established PLCs can help teachers increase disciplinary knowledge, enhance teaching skills and develop professional identity [8–10]. The second category includes the factors contributing to the operation of PLCs. For example, according to Hord et al. [11], teachers' availability is a key factor in ensuring the regular organization of PLCs. Mulford [12] and Robison et al. [13] identify that PLC leadership is essential for the effective implementation of PLC activities. Hall and Hord [14] argue that supportive policies outside the schools are also important for the organization of PLCs.

PLCs also exist in China, although there are some variations in their naming and implementation. The PLCs organized for teachers in Chinese K-12 schools are called *jiaoyanzu* (Teacher Research Groups in English, hereafter TRGs). Rooted in China's collectivist culture, which emphasizes harmony, authority and unity [15], TRGs have distinctive Chinese characteristics in terms of their organization and function. In terms of organization, they are grassroots teaching research units that are established according to the different subjects in schools [16]. With regard to function, TRGs focus on "teaching", whereas PLCs in Western countries are more concerned with "learning" [16]. Despite their distinctive Chinese characteristics, TRGs have many similarities to PLCs in Western countries [17]. For example, they both emphasize the importance of shared vision, collective responsibility for improving student learning outcomes and collaborative culture among teachers [18]. Similar to PLCs in form, TRG activities involve joint lesson planning, sharing of materials and ideas, peer observation and mentoring, collective interrogation of open lessons and conducting collaborative research [16,19,20]; in function, TRGs contribute significantly to teacher learning and professional development [21], while serving as a crucial platform for teachers to collaborate with each other to tackle the common problems encountered in classroom teaching [22,23]. Therefore, TRGs are regarded as the Chinese model of PLCs. When Shanghai topped the Programme for International Student Assessment, many scholars attributed its success to the implementation of TRGs in China. The Chinese model of PLCs, represented by TRGs, has thus been attracting increasing attention from international scholars. A brief literature review on the Chinese model of PLCs revealed that the existing research focuses on their historical development [24], organizational form [25] and function with respect to teacher professional development [26]. Overall, previous studies of the operation of PLCs have primarily been conducted in Western countries, whereas those in the Chinese context are limited [24]. Moreover, research on PLCs in the Chinese context is often based on single subjects, as PLCs have traditionally been established according to the different subjects in schools [27–29].



*1.2. The Chinese Context*

Over the past few decades, educational philosophy has undergone a radical change around the world, shifting from disciplinary learning to interdisciplinary learning [30]. Accordingly, enhancing students' interdisciplinary competences to succeed in an uncertain and constantly changing era has become the primary mission of schools, colleges and universities worldwide. In response to this challenge, in 2014, the Ministry of Education of China issued Opinions on Comprehensively Deepening Curriculum Reform and Implementing the Fundamental Tasks of Moral Education, highlighting the importance of interdisciplinary cooperation [31]. In 2019, the Central Committee of the Communist Party of China and the State Council of China released China's Education Modernization Plan Towards 2035 to guide the education development in the country. According to this plan, the Chinese government will "promote the interdisciplinary integration in primary and secondary schools and explore the development of interdisciplinary curricula focusing on cultivating students' comprehensive quality" [32]. To facilitate the development of interdisciplinary curricula and enhance teachers' capacity to deliver interdisciplinary curricula, Chinese K-12 schools have been enthusiastic about establishing PLCs in interdisciplinary subjects.

Although many PLCs in interdisciplinary subjects have been established in China, only a few articles have been published about them. These articles illustrate what PLCs in interdisciplinary subjects are in the Chinese context and how PLCs in interdisciplinary subjects benefit teacher professional development in China. According to Chen [33], PLCs in interdisciplinary subjects in China refer to teaching and research teams composed of teachers from different subjects and grades. Tang [34] and Zhao and Chen [35] reveal that PLCs in interdisciplinary subjects provide support for cooperation between teachers from different subjects, broaden teachers' horizons and improve the development of professional learning. It is notable that all these articles are written in Chinese and are inaccessible to the wider international research community. Moreover, little has been mentioned about the factors impacting the sustainable operation of the PLCs in interdisciplinary subjects, although this emerging model of PLCs is considered to be an important platform for teacher professional development in China. Questions such as what factors impact the sustainable development of PLCs in interdisciplinary subjects, how they promote teacher professional development and what the differences in factors impacting PLCs in interdisciplinary subjects are, as well as those impacting the traditional subject based PLCs, remain unanswered. In this context, to fill this research gap, we attempt to answer the following research questions in this study: (1) What factors are impacting the sustainable development of PLCs in interdisciplinary subjects in Chinese K-12 Schools? (2) How are the factors impacting the development of PLCs in interdisciplinary subjects different from those impacting the development of traditional subject-based PLCs? By answering these two research questions, we can understand the systematic and positive effect of PLCs in interdisciplinary subjects on teacher professional development, which further influences student achievement and school improvement.

The paper is organized as follows. It starts with a description of the analytical framework and then sets out the research methodology. The research findings are subsequently presented and discussed on the basis of the data collected. The conclusion summarizes the significance of this study and provides directions for future research.

## 2. Analytical Framework

This study adopted the Global PLC conceptual framework as the analytical framework. The Global PLC conceptual framework was constructed by a research team led by Huffman [36] on the basis of Senge's [37] five disciplines of learning organizations and Hord's [5] model of PLCs. In the 1970s, in the face of the fiercely competitive market, the concept of "organizational learning" [38] was introduced to describe a situation wherein companies could only maintain their competitive advantage through continuous learning. As the pioneer of research on organizational learning, the Massachusetts Institute of Tech-

nology established the Center for Organizational Learning in 1990. While working at the Centre for Organizational Learning, Senge developed five disciplines of learning organizations: systems thinking, personal mastery, mental models, building a shared vision and team learning [37]. Later, Hord, an educational researcher working on school leadership and school improvement, had the opportunity to work in a "learning organization" to investigate how a school improved through PLC activities. Inspired by Senge's five disciplines of learning organizations, in 1997, Hord [5] proposed a model of PLCs in education, which was composed of shared values, vision and goals, collective learning and application, shared individual practice, shared and supportive leadership and supportive conditions. Hord argued that this model could contribute to school improvement and education reform. Since it was proposed, Hord's model of PLCs has been very well received by educational researchers in the Western world and has been validated and revised to adapt it to different cultural contexts.

Based on Hord's model of PLCs, Huffman et al. [36] developed the Global PLC conceptual framework in 2009 by conducting a literature review of PLC actions in schools in mainland China, Hong Kong, Taiwan, Singapore and the United States and interviewing principals and teachers in these five countries and regions. Having noticed common practices among school improvement initiatives, Huffman and her colleagues identified five constructs impacting PLC actions: organizational structure, policies and procedures; leadership; professionalism; learning capacity; and sense of community. Organization structures, policies and procedures are considered to be structures which constitute an important form of support for teacher learning in PLCs. The other four constructs are considered as actions and refer to behaviors, relationships and interactions among teachers. Leadership can be considered to be both a structure and an action within schools. These five constructs are considered as important internal factors for teachers in carrying out PLC actions, but given the cultural contexts of different countries and regions, external factors, such as the central office, state/province/city and parents/family, have also been identified as influencing PLC actions [36].

The first construct, organizational structures, policies and procedures, provides supportive conditions in the development of PLCs. The key descriptors include time, scheduling, space, staffing, funding, equipment/technology, policies at the national, local and school levels and monitoring and assessment practices that impact PLC development. The second construct is leadership, which refers to the key strategies school leaders adopt to model, facilitate, develop responsibility for and foster shared values, visions and decision making among teachers in the school. The key descriptors of this construct are modeling learning practices, facilitating learning for all, sharing a vision, broad-based decision making, moral purpose and responsibility. The third construct is professionalism, which represents actions related to excellence in which teachers act responsibly and are committed to colleagues and students, to adopting best practices and to demonstrating ethical behavior and good judgment. The key descriptors involve teachers' commitment, identity, ethics, professional judgement and self-efficacy. The fourth construct is learning capacity, which emphasizes teachers' openness and capacity to learn new strategies, to use data and to adjust to changes in practice. The key descriptors include teachers' openness to learn collaboratively, their ability to learn through collective inquiry, to use multiple sources of data, to give supportive feedback to colleagues, to reflect on their colleagues' feedback, to improve teaching styles and collective efficacy. The fifth construct is the sense of community, which refers to a school culture and atmosphere focusing on the relational aspect of the community to ensure high levels of learning for teachers and students. The key descriptors cover shared values and norms, mutual trust and respect, collegial influence, group membership, emotional bonding and recognition and celebration.

The Global PLC conceptual framework identifies several factors impacting PLC actions in five countries and regions with different cultural backgrounds. The identified factors provide essential support for the sustainable development of PLC actions. Therefore, this study will adopt the Global PLC conceptual framework to explore the factors impacting

the sustainable development of PLCs in interdisciplinary subjects such as Education for International Understanding in Chinese K-12 schools. The empirical investigation will test this conceptual framework's applicability in PLCs in interdisciplinary subjects and modify it if needed.

## 3. Research Methodology

To obtain an in-depth understanding of the factors impacting the sustainable development of PLCs in interdisciplinary subjects in Chinese K-12 schools, this study used the Global PLCs conceptual framework proposed by Huffman et al. [36] as the analytical framework and adopted a case study as the research methodology. According to Yin [39], a case study should be performed when the research purpose is to "investigate a contemporary phenomenon in depth and within its real-life context, especially when the boundaries between phenomenon and context are not clearly evident". In this section, we give a brief introduction of the selected case and the data collection, analysis and validation procedures.

### 3.1. Case Selection

This study selected Chengdu Horsens Primary School as the case. Chengdu Horsens Primary School is a public elementary school founded by the Chengdu Government in China in cooperation with the Government of Horsens in Denmark in 2018. It is a member of Chengdu's Education for International Understanding (hereafter EIU) Studio. The subject of EIU in Chengdu Horsens Primary School is mainly composed of interdisciplinary knowledge aiming to promote cross-cultural exchange and the understanding of different countries and is complemented by other subjects and theme activities infused with relevant knowledge and skills. The interdisciplinary EIU subject in this school is run for students at all grade levels and is taught by both full-time EIU teachers and English teachers. Teachers of other subjects, such as Chinese, Mathematics and Fine Arts, are encouraged to infuse knowledge on international understanding into their subjects and to offer inquiry-based theme activities. Moreover, the principal has established the EIU Curriculum Studio to promote curriculum development and teacher training in this school and beyond. To engage more teachers in the curriculum development of EIU, Chengdu Horsens Primary School has established the "1 + 3 + N" PLC model, which is characterized by interdisciplinarity. In this model, 1 refers to a seed teacher responsible for the development of each EIU course; 3 represents three teachers of any subject, such as Chinese, English, mathematics, fine arts, physical education and moral education; and N is any teacher who is willing to participate in the curriculum development. The one seed teacher and three teachers of any subject are key members of the PLCs. The PLC model implemented in Chengdu Horsens Primary School is designed to stimulate teachers to demonstrate their teaching in PLCs and therefore facilitate the curriculum development of EIU. As of now, this school has established six "1 + 3 + N" PLCs.

The reason for selecting Chengdu Horsens Elementary School's PLCs in EIU as the case is because this school has developed an exemplary PLC model in interdisciplinary subjects. Through participating in the PLC activities, teachers of other subjects accumulate plenty of knowledge and experience in the curriculum development and teaching practices of EIU. They also improve their interdisciplinary competences and enhance their teaching capacity in their own subjects. Therefore, Chengdu Horsens Elementary School's experiences are worth exploration and dissemination.

### 3.2. Data Collection

As part of this study, semi-structured interviews were conducted to explore factors impacting the sustainable development of PLCs in interdisciplinary subjects in Chinese K-12 schools. In total, we recruited one principal, one course director, five seed teachers and 13 teachers involved in the PLCs to participate in this study. Table 1 presents the profile of the participants and our coding system. Participants in different positions were interviewed in order to be able to draw a complete picture of how PLCs in EIU operate in Chengdu

Horsens Elementary School. Our interview questions mainly involved three aspects. The first set of questions was asked to obtain the participants' background information, such as their motivation in teaching EIU and basic information on the course they were teaching. The second set of questions was designed according to the Global PLC conceptual framework constructed by Huffman and her colleagues. The questions focused on the five constructs of organizational structure, policies and procedures; leadership; professionalism; learning capacity; and sense of community. The third set of questions was proposed in order to understand the similarities and differences between PLCs in EIU and PLCs in other subjects. Examples of the interview questions are presented below.

- What is your motivation for teaching for EIU? Please introduce basic information on the course you are teaching.
- What do you think are the main factors impacting the implementation of PLC activities for the subject of EIU in your school? (Hints: time, space, staffing, funding, equipment and other resources).
- What learning capacities do you think are essential for facilitating PLC activities in EIU? (Hints: openness to learn collaboratively, ability to inquire collectively, ability to use multiple sources of data, ability to provide feedback, ability to reflect on and change practices and sense of collective efficacy).
- What are the similarities and differences between PLCs in EIU and PLCs in other subjects?
- How have the PLC activities in EIU changed your teaching of the subject and of your own subject (Chinese, English, Mathematics, etc.)?

**Table 1.** Numbers and coding system of participants by position.

| Positions in the Case School | Number of Interviewees | Coding System |
|---|---|---|
| Principal | 1 | A1 |
| Administrator | 1 | B1 |
| Seed teacher of the PLCs * | 5 | C1–C5 |
| Teacher participating in the PLCs ** | 13 | D1–D13 |
| Total | 20 | |

* Seed teachers of the PLCs are composed of two full-time EIU teachers and three teachers teaching the subjects of Chinese, English and Fine Arts. ** Teachers included one teacher of Chinese, one teacher of Physical Education, one teacher of Moral Education, two teachers of Mathematics, two teachers of Fine Arts and three teachers of English.

In addition to the questions listed above, the interviewer also asked some follow-up questions wherever appropriate. For example, when the participants mentioned that the PLCs in EIU had a positive impact on their teaching in their own subject, the interviewer asked about the specific aspects of the positive impact. The interviews were conducted individually. Each interview lasted about 40 min and was recorded. We also observed the operation of the PLCs in EIU and recorded the teachers' emotional responses and body language while participating in the PLC activities. The observations were performed three times for a total of 10 h. Additionally, we collected teachers' journal entries about their teaching reflections.

### 3.3. Analysis and Validation

Upon data collection, the first author and the third author transcribed the data and conducted a thematic analysis by following Braun and Clarke's [40] six-phase guide. The themes were identified through a combination of inductive and deductive approaches. The preliminary coding scheme was first designed through a deductive approach according to the Global PLC conceptual framework. The five constructs in the Global PLC conceptual framework, i.e., organizational structure, policies and procedures, leadership, professionalism, learning capacity and sense of community, were identified as parent nodes. The key descriptors in each of the five constructs, such as group membership, collegial influ-

ence and emotional bonding in the construct of sense of community, were coded as child nodes. Later, the first author and the third author revised the preliminary coding scheme through an inductive approach. They repeatedly read the transcriptions and formulated a formal coding scheme based on the data collected (see details in Figure 1). For example, when constructing the Global PLC conceptual framework, Huffman et al. [36] identified professionalism and learning capacity as two separate constructs. According to their interpretation, professionalism is concerned with teachers' professional competence, whereas learning capacity stresses teachers' lifelong learning and self-improvement. In this study, however, the first author and the third author found it difficult to distinguish these two constructs, so they put them together in one parent node in the data analysis process.

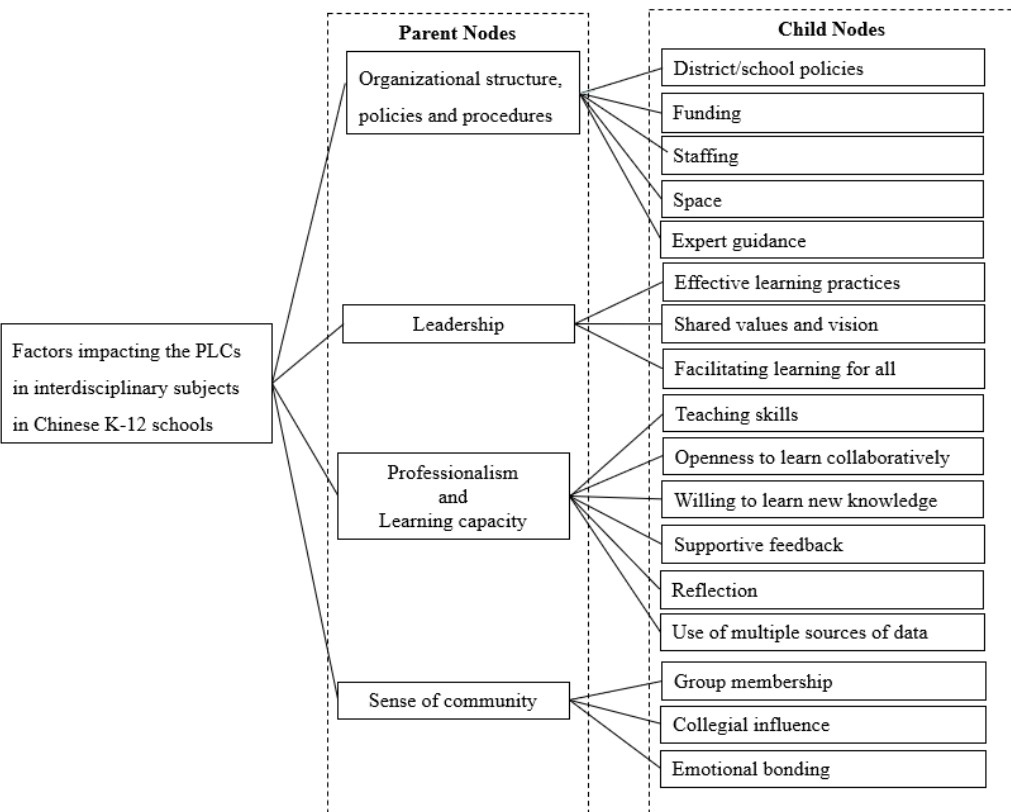

**Figure 1.** The coding scheme of factors impacting the sustainable development of PLCs in interdisciplinary subjects in Chinese K-12 schools.

After the formal coding scheme was formulated, the first author and the third author coded the data independently in a qualitative data analysis software tool N-vivo 12.0. When all the transcriptions had been coded, they conducted a coding comparison query in NVivo to assess the inter-rater reliability. The Kappa coefficient across all nodes was reported to be 0.76, indicating a substantial agreement between the two authors. In the end, the two authors discussed the disagreements and revised the discrepancy codes until consensus was reached. To examine the validity and reliability of the coding, the first author and the third author triangulated the data collected from different participants and from different data sources to ensure that they did not conflict with each other. In addition, the paper was shared with interviewees to allow them to clarify or change the data interpretations.

## 4. Results

In this section, we present our analysis of the factors impacting the sustainable development of PLCs in interdisciplinary subjects in Chinese K-12 schools. Our analysis was guided by the Global PLC conceptual framework. Since both professionalism and learning

capacity in the Global PLC conceptual framework point to teacher capacity development and they cannot be completely differentiated, we present them together below.

### 4.1. Organizational Structure, Policies and Procedures

The interviews demonstrate that the PLCs in EIU in Chengdu Horsens Primary School have been strongly supported by the local government and the school. In 2019, the Education Development Centre of Chengdu High-Tech Industrial Development Zone (hereafter the Zone), where Chengdu Horsens Primary School is located, signed a cooperative agreement with a university in Beijing to develop the project "Joint Construction of National Demonstration Pilot Zone for the Internationalization of Basic Education". As a part of this project, the Zone selected several pilot schools within the jurisdiction to offer the interdisciplinary subject of EIU. Chengdu Horsens Primary School was selected as a pilot school in 2020 and has received consistent support from the Zone and the school in funding, staffing, space and expert guidance since then. These supports provide a solid foundation for the sustainable development of the PLCs.

> *The Chengdu High-Tech Industrial Development Zone has provided strong support for the pilot schools. Our school has not only received generous funding from the Zone to establish the PLCs in EIU, to develop EIU curriculum and to construct relevant facilities, but also received guidance from experts in Beijing. Through the cooperation with a university in Beijing, the Zone invites educational experts to provide guidance to each pilot school, including providing advice on the PLCs' organization.* (A1)

Thanks to the policy and funding support from the Zone and the school, PLCs in EIU were very quickly established in Chengdu Horsens Primary School. Unlike the traditional subject-based PLCs in China, which are composed of teachers in the same grade and the same subject [18], PLCs in EIU prioritize interdisciplinarity. Chengdu Horsens Primary School has thus established the "1 + 3 + N" PLC model, engaging teachers from many different subjects. This new PLC model provides institutional support for teachers to exchange experiences and promote mutual learning in EIU. These PLCs are nonmandatory and open to every teacher in the school.

> *Each semester, the school establishes three PLCs in EIU, composed of both designated teachers and voluntary teachers. Teachers are also encouraged to establish PLCs by themselves and to submit lesson plans for the school to select.* (B1)

In addition to the support in policy, funding, staffing and expert guidance, Chengdu Horsens Primary School provides a meeting room for PLCs in EIU, ensuring that the teachers have the space to discuss and learn in groups.

> *Our school has built a meeting room specialized for the PLCs in EIU and built a cultural wall in the room to promote EIU. EIU has been identified as a featured subject in our school, so we can apply for funding for this subject more easily. Our group usually meets once a week to develop the curriculum. But we meet more often when there is a teaching competition on EIU at the school or the Zone level. The number of our meeting also increases when experts from the university in Beijing come to provide guidance.* (C1)

### 4.2. Leadership

Leadership is crucial to the successful implementation of a school project [41]. In the Global PLC conceptual framework, leadership is considered to be both a structure and an action [36]. Resonating with this argument, the principal of Chengdu Horsens Primary School is both an important leader and a significant actor in the PLCs in EIU. The principal has not only been leading the development of EIU, including the operation of PLCs, but has also been working as a full-time EIU teacher in the school. Owing to her previous work experience in foreign exchange and cooperation in basic education in government, she prioritized the cultivation of students' cross-cultural competence and encouraged teachers to participate in PLCs in EIU.

*We were anticipating the outlook of the school when it was founded. Considering the long-lasting cooperation between Chengdu and Horsens, we decided to connect the international resources of both cities and develop the curriculum for EIU. Before heading this school, I worked at the Department of Education on foreign exchange and cooperation for a long period. The previous work experience cultivated in me a keen sense of internationalization. Once the school was founded, we applied for the EIU program launched by the Chengdu High-Tech Industrial Development Zone in cooperation with a university in Beijing, but we failed to be included in the first batch of the pilot schools due to an insufficient number of teachers and students. Fortunately, we participated in the construction of the first batch of pilot schools as an observer and were selected as part of the second batch of polit schools a year later.* (A1)

The principal's focus on the PLCs in EIU has impacted every teacher in Chengdu Horsens Primary School. Her impact has been mainly exerted in three ways. First, the principal established the "1 + 3 + N" PLC model to stimulate teachers' interests in teaching the interdisciplinary of EIU and provide a platform for effective learning practices. Second, the principal developed the EIU database together with colleagues to offer teaching materials in EIU, so that other teachers participating in the PLCs are able to update and expand the database. In this way, she has helped construct the school culture promoting EIU and established shared values and vision among teachers. Third, the principal established an EIU Curriculum Studio, which, as a form of PLC, provides a platform for exchanges and mutual learning to teachers in the school and beyond, thus facilitating teachers' learning.

*In PLC activities, the most important things for us are to polish the lesson plans and develop the teaching materials. The school has built an EIU database, which includes teaching materials developed by three full-time EIU teachers in our school. As a growing number of teachers become involved in the PLCs in EIU, the teachers can improve their courses on the basis of the teaching materials provided in the EIU database. They can also upload their courses to provide teaching materials to other teachers. In addition, many teachers have developed new EIU courses and PLC activities and uploaded them to this database for other teachers in this school to learn from and improve.* (D5)

*The studio has a three-year plan. The task for the first year, which is 2021, is to complete the organization of the studio and formulate relevant regulations. The studio has 29 teachers, including 18 from the Zone and 11 from outside the Zone and this number is expected to increase in the coming years. In the second year, we will strengthen the training of EIU teachers, attempting to promote the development of EIU in other schools. In the third year, we will promote our EIU curricula in rural schools.* (A1)

### 4.3. Professionalism and Learning Capacity

When constructing the Global PLC conceptual framework, Huffman et al. [36] identified professionalism and learning capacity as two constructs. According to their interpretation, professionalism is concerned with teachers' professional competence, whereas learning capacity stresses teachers' lifelong learning and self-improvement. This study finds it difficult to distinguish these two constructs, so we present findings related to them together in this subsection. The interviews demonstrate that, compared to traditional subjects, the interdisciplinary subject of EIU sets higher requirements on teachers, requiring them to be open, collaborative and reflective, to have interdisciplinary competences, to have the willingness and ability to learn new knowledge and to be able to improve their teaching based on their colleagues' feedback. Since EIU is an emerging subject in China, there is no programs or schools specialized for the cultivation and training of EIU teachers. Currently, teachers teaching EIU courses in China are those teaching traditional subjects such as Chinese, English and Mathematics. During the interviews, the participants repeatedly mentioned the criteria for selecting EIU teachers. First, the teachers should possess proficient teaching skills.

*We pay particular attention to teaching skills when selecting teachers for the subject of EIU. We find that teachers with proficient teaching skills in their own subjects are likely to teach EIU with ease.* (B1)

Second, the teachers must have interdisciplinary competences and openness to learn collaboratively. EIU is not confined to a single subject, so the teachers' ability to integrate different subjects is essential. The PLCs in EIU provide teachers in different subjects with a unique platform to develop interdisciplinary competences, while strengthening their self-motivation and inspiring their self-development.

*After quitting my job as an English teacher, I started to teach EIU for Grade Three as a full-time EIU teacher in this school. In one EIU course, to share the understanding of death, I compared the Day of the Dead in the West with the Qingming Festival in China. When preparing for the course, I encountered trouble in finding features about the Qingming Festival. Fortunately, the teacher in the subject of Chinese in our PLC did me a favor and helped me find relevant materials.* (C3)

*I am an English teacher and I am used to teaching in English, so it is really hard for me to make EIU students understand the teaching contents in concise Chinese words. The teacher in the subject of Chinese in our PLC observed my classroom teaching several times and helped me improve the teaching language.* (C4)

Third, the teachers should be willing to develop new knowledge, to be reflective and to break their own boundaries. Teachers' teaching quality is closely related to the update of teaching materials, their teaching research and reflections on teaching feedback. As previously mentioned, there are no well-developed teaching materials for EIU. Hence, the teachers in PLCs in Chengdu Horsens Primary School should be more capable of the teaching material preparation than those in traditional subject-based PLCs.

*I have taught the subject of Chinese for six years and get tired of delivering the same teaching content every day. My enthusiasm for teaching was ignited when I started to participate in PLC in EIU, as I was exposed to different ideas by working with teachers from different subjects. Moreover, my experience in PLC in EIU has positively impacted my teaching in the subject of Chinese. When teaching Chinese, I try to integrate international elements into my class in order to enhance students' international understanding.* (C5)

*We have attended many conferences on EIU at the Zone, district and national levels. The EIU teaching demonstrated by experts and teachers outside our school provides excellent materials for our curriculum. After joining in the PLC in EIU, I started to collect teaching materials related to EIU, particularly through Weibo, WeChat, the Red Book and shared them with other teachers at the PLC activities. We subsequently integrated them into the teaching materials.* (C2)

*4.4. Sense of Community*

A sense of community, which is highly dependent on the school's culture and climate, provides a supportive condition for teacher professional development [36]. The interviews reveal that the sense of community in PLCs in EIU in Chengdu Horsens Primary School mainly consists of three components: group membership, collegial influence and emotional bonding. Group membership is the basic unit of a community. Members in the same community share common goals, comply with common requirements and gradually develop a sense of belonging and security while interacting at the PLCs. The participants emphasized that "as the number of PLCs in EIU continues to grow, teachers' sense of community also grows" (D2 and D4). From this quotation, we can find that PLCs and teachers participating in PLCs influence each other. The teachers' improvement in PLC activities can increase the PLCs' cohesion and, in turn, PLCs with greater cohesion and influence can attract more teachers to get involved.

*I joined the PLC in EIU because my colleagues had participated in it. The PLCs made up of teachers from different subjects have enabled me to find out about other teachers'*

*perspectives. Additionally, teachers in the PLCs are highly motivated, as evidenced by the fact that they would like to attend PLC activities after finishing classes and handling routine work. I see their real passion for EIU. Now, I am planning to apply for one EIU course next semester to improve my teaching skills.* (D5)

The shared experiences (A1), shared time (C4), shared gains (C5) and good relationships among teachers constitute the unique emotional bonding in the PLCs and shape the collective identity of teachers participating in them. Since nobody has any teaching experiences in the interdisciplinary subject of EIU, there is no distinction between novice and veteran teachers in this subject. Therefore, PLCs in EIU create a more equal atmosphere than those in traditional subjects.

*In the PLC activities of traditional subjects, we are afraid to speak out our ideas because of the presence of veteran teachers. Yet in PLC activities in EIU, no one has the authority, so the teachers feel free to express their opinions. In this atmosphere, we are able to communicate and brainstorm on an equal basis.* (C4)

## 5. Discussion

By adopting the Global PLC conceptual framework as the analytical framework, this study identified factors impacting the sustainable development of PLCs in interdisciplinary subjects in Chinese K-12 schools. On the basis of the research results, it can be found that PLCs in interdisciplinary subjects, which are significantly different from PLCs in traditional subjects, offer an excellent platform for teachers in different subjects to collaborate with each other. They can more effectively improve teachers' self-efficacy, foster teachers' emotional bonding and promote teacher professional development. The Global PLC conceptual framework constructed by Huffman et al. [36] is validated in this study, proving that school structures and policies, school leadership, teachers' professionalism and learning capacity and their sense of community are major factors impacting the sustainable development of PLCs in interdisciplinary subjects in Chinese K-12 schools.

First, well-developed school structures and policies provide institutional support for the operation of PLCs in EIU in Chengdu Horsens Primary School. As of now, the curriculum development of EIU has received strong support from the local government and the school. The school, as the major force promoting EIU, has adopted various strategies to facilitate the operation of PLCs in EIU, such as providing funding and space for PLC activities, inviting university experts to provide guidance for PLCs, establishing an online database for resource sharing and fostering the cultural construction on international understanding. This finding confirms Hord's [5] argument that schools' supportive conditions provide the facilities and necessities of when, where and how the staff can come together as a unit to learn, make decisions and carry out new practices.

Second, the principal of Chengdu Horsens Primary School has played a leading role in the PLC activities in EIU. Good leaders can develop visions for their schools according to their personal and professional values. They tend to seize every opportunity to express their goals and visions and thus influence their teachers and other stakeholders. Normally, their schools' educational philosophy, organizational structures and activities are also developed on the basis of their visions [42]. In this study, by prioritizing the interdisciplinary subject of EIU in the school, the principal of Chengdu Horsens Primary School has promoted the operation of PLCs in EIU. Moreover, she has built an EIU Curriculum Studio for teachers in the school and beyond to promote information exchange on EIU.

Third, the subject of EIU requires teachers to have proficient teaching skills and have the ability to integrate different subjects, to learn new knowledge and to reflect on their own teaching. Among these requirements, proficient teaching skills are most critical for becoming a qualified EIU teacher. Moreover, teachers need to have the ability to integrate knowledge from different subjects. In contrast with traditional subjects, the interdisciplinary subject of EIU requires teachers to mobilize knowledge from different subjects and to communicate with and learn from teachers in other subjects. In addition, they need to break down the boundaries between different subjects, foster new knowledge

and reflect on the feedback offered by their colleagues, with the aim of promoting their professional development.

Fourth, with the growing number of teachers participating in PLCs in EIU, their communication and mutual influence may also increase, contributing to the construction of the sense of community. The community culture of PLCs in EIU is characterized by a high degree of openness and equality. Compared to the traditional subject-based PLCs, PLCs in EIUs are nonmandatory. Due to the institutional support and the peer influence among teachers, PLCs in EIU grow very fast, leading to the formation of a school culture promoting the development of EIU. In addition, since every teacher teaching the subject of EIU is a novice, there is no "authority" in the PLCs. Hence, the teachers can trust each other and are able to share their experiences more equally.

In addition, this study identified three differences between PLCs in interdisciplinary subjects and those in traditional subjects. The first difference is that PLCs in interdisciplinary subjects place more emphasis on a collaborative atmosphere inclusive of both harmony and conflict. Lauer and Dean [43] reveal that mutual support and cooperation among teachers is indispensable in the operation of PLCs. Therefore, a harmonious and cooperative atmosphere plays an important role in the sustainable development of PLCs. In practice, a harmonious atmosphere in PLCs is indeed crucial for promoting teacher professional development. However, an excessive pursuit of harmony in the community may lead to teachers' reluctance to challenge authority, reduce their passion for innovation and decrease their ability to make changes, which in turn inhibits the sustainable development of PLCs [27]. In traditional subject-based PLCs, teachers pursue harmonious relationships with their colleagues and are unwilling to challenge more authoritative members. They tend to avoid conflicts and confrontations, which makes critical discussions difficult. Therefore, many problems cannot be easily resolved [18]. In this case, different opinions and teaching methods are also suppressed [44], making teacher professional development hard to achieve. However, in interdisciplinary PLCs, teachers come from different subjects and different grades. There are no authorities in such PLCs. Hence, they do not need to worry about violating the authorities or hurting relationships with other colleagues. Although the teachers participating in PLCs in EIU have conflicting opinions, they are not afraid to share different opinions. They believe that these conflicts enable them to achieve continuous improvement. The exchange of opinions and the conflicts in PLC activities empower them to achieve professional development. After all, conflicts are also crucial in social interactions [45]. On the basis of the research results, it turns out that a cooperative atmosphere in which harmony and conflict coexist is more conducive to the effective operation of PLCs in interdisciplinary subjects.

Second, PLCs in interdisciplinary subjects are dedicated to pursuing the coexistence of individual visions and shared visions rather than only pursuing the shared vision of the community. One characteristic of PLCs is that they have shared values, visions and goals [46]. While engaging in PLCs, teachers are influenced by the values, norms and practices prevailing in the PLCs and thus develop a shared vision [47]. In traditional subject-based PLCs, most teachers focus on the shared vision while neglecting their individual visions. However, it has been recognized that placing individual visions under the shared vision is a potential "danger" for PLCs' sustainable development [47]. When community members hide their true feelings, their motivation to achieve the shared vision will greatly decrease. In addition, they may not proactively pursue progress and growth within the community. In this case, they have to hide their real feelings, which may decrease their motivation to pursue improvement [48]. In Chengdu Horsens Primary School, since the teachers participating in PLCs in EIU have different subject backgrounds, their roles in PLCs are differentiated. The PLCs in EIU thus create a space where individual visions and shared visions can develop together. The PLCs in EIU in Chengdu Horsens Primary School are characterized by openness, as every teacher is allowed to apply to establish PLCs in EIU and the PLCs are open to every teacher in the school. Therefore, the shared visions of the PLCs in EIU are produced based on the continuous exchange and integration of

individual visions. The exchange and integration of individual visions inspire teachers to improve learning autonomy and teaching ability in their own subjects. Consequently, the professionalism of the teachers is enhanced and the sustainable development of PLCs is achieved.

Third, with institutional support from the local government and the school, PLCs in interdisciplinary subjects focus more on an emotional bonding identity. Institutional support provides the infrastructure and requirements of when, where and how the staff can come together as one unit to learn, make decisions and implement new practices [49]. This is necessary for the long-term development of PLCs, because PLCs built on personal relationships are not stable [50]. To avoid instability, many organizations offer institutional support for PLCs to maintain their sustainable development. Meanwhile, organizational leaders are willing to provide institutional support for teachers to collaborate with each other. In practice, institutional support can promote teacher professional development in the beginning of the PLCs' establishment. However, relying solely on institutional support while neglecting teachers' internal development needs will lead to the construction of PLCs becoming formalistic [51], particularly when the administrative interference in PLC activities is excessive. After all, some PLC activities are implemented to fulfill orders from the educational administration, rather than to promote teacher professional development [52]. In this regard, PLCs must strengthen their identity by creating emotional links. It is obvious that teachers participating in PLCs in Chengdu Horsens Primary School are able to fully express their feelings, communicate with each other and promote professional development together, thus forming tightly connected communities linked by emotions. These PLCs continue to expand and continue to pursue a more sustainable symbiotic environment by attracting more teachers to join them. From the above discussion, it can be inferred that the basis for promoting PLCs' sustainable development should be the identity relationship established by the emotional link [48].

In sum, this study argues that the five factors of organizational structure, policies and procedures, leadership, professionalism, learning capacity and sense of community can promote the sustainable development of PLCs in interdisciplinary subjects in Chinese K-12 schools. Compared to traditional subject-based PLCs, PLCs in interdisciplinary subjects particularly highlight the value of sense of community in their development process and present three distinctive features around this: a conflict-inclusive atmosphere, the coexistence of individual and shared visions and an emotional bonding identity. These three features have helped address the problems existing in traditional subject-based PLCs in Chinese K-12 schools to a certain extent. In the interdisciplinary PLCs, teachers' individual visions are respected and their motivations for self-growth and self-development are stimulated. Teachers are more proactive in participating in PLC activities and in communicating with teachers in other subjects, which is more conducive to promoting teacher professional development and to enhancing teaching quality.

## 6. Conclusions

Based on the empirical investigation of PLCs in EIU in Chengdu Horsens Primary School, this study explored the factors impacting the sustainable development of PLCs in interdisciplinary subjects in Chinese K-12 schools. These research results make significant contributions to the field of PLCs. On the one hand, this study confirms the influence of organizational structure, policies and procedures; leadership; professionalism; learning capacity; and sense of community on the sustainable development of PLCs in interdisciplinary subjects. On the other hand, this study distinguishes factors impacting the development of PLCs in interdisciplinary subjects from those impacting the development of the traditional subject-based PLCs in China. We highlight three differences, which include a conflict-inclusive atmosphere, the coexistence of individual and shared visions and an emotional bonding identity. These three distinctive features also contribute to the sustainable development of PLCs in interdisciplinary subjects in Chinese K-12 schools. It should be noted that PLCs are embedded in cultures [53]. Hence, the findings generated

in this study should not be directly transplanted to other cultural contexts. Nevertheless, the study of PLCs in interdisciplinary subjects in Chinese K-12 schools promotes the understanding of the operation of PLCs in China as most of the existing studies focus on the traditional subject-based PLCs in China. The study also provides implications on the implementation of the PLCs in interdisciplinary subjects for other K-12 schools in China and on the improvement of PLCs in other cultural contexts.

In addition, we are aware that this study has some limitations. The foremost is that it was a single case study and did not carry out ongoing research on the case's PLC activities. Therefore, we advise future scholars to conduct more research on the PLCs in interdisciplinary subjects in other Chinese K-12 schools and conduct a comparative study of the PLCs in interdisciplinary subjects in different regions of China with the aim of identifying their common characteristics. They are also advised to identify the factors that promote and hinder teacher professional development in PLCs in interdisciplinary subjects in Chinese K-12 schools. In addition, we advise future scholars to conduct ongoing research on other PLCs in interdisciplinary subjects in Chinese K-12 schools to explore their operation and relevant issues in order to further promote teacher professional development in China and across the world.

**Author Contributions:** Conceptualization, Y.H. and X.J.; data collection and analysis, Y.H. and Y.Y.; writing—original draft preparation, Y.H.; writing—review and editing, X.J. All authors have read and agreed to the published version of the manuscript.

**Funding:** This research was funded by the Beijing Social Science Planning Project, "A comparative study of the education as a soft power tool in international metropolises", grant number 21JYC014.

**Institutional Review Board Statement:** The study was approved by the Institutional Review Board of Beijing Normal University (Protocol code: BNU202104100020; Date of approval: 11 May 2021).

**Informed Consent Statement:** Informed consent was obtained from all participants involved in the study. Written informed consent has been obtained from the participants to publish this paper.

**Data Availability Statement:** The datasets presented in this article are not readily available because original data contains personal information of the research participants. Requests to access the datasets should be directed to jxltxszh@126.com.

**Acknowledgments:** The authors would like to thank all participants for contributing their time and effort to this study. In addition, we would also like to thank the reviewers of this paper for valuable comments.

**Conflicts of Interest:** The authors declare no conflict of Interest.

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
