# Peer review of "Factors Impacting the Sustainable Development of Professional Learning Communities in Interdisciplinary Subjects in Chinese K-12 Schools: A Case Study"

_sustainability, doi:10.3390/su142113847_

Round 1
Reviewer 1 Report
Comments and suggestions about the paper:
- The article titled "Factors Impacting the Sustainable Development of Professional Learning Communities in Interdisciplinary Subjects in Chinese K-12 Schools: A Case Study" is within the scope of a current topic.
- This paper was a very interesting, and it is well structured and the language is scientific.
- Regarding the method, how was the data analyzed? Please explain the analysis phases.
- What is the analysis technique? Please explain this as part of your method.
- Can you explain how the coding was done – how many coders did you have? Who were they? What was the coding method?
- The authors present the research as qualitatively, but there is no triangulation of data, a characteristic of qualitative research.
- Have the authors considered the theoretical framework or any theories when conducting the research?
- The scientific relevance of the study is not stated (What is the gap in the literature that is addressed by this study?)
- The "Future Research Directions" section of the research is missing.
- Have a deeper investigation on the data, since the current text is purely descriptive and have been well documented in the existing literature. It's not new.
- There is no discussion on the novelty that the study would like to add in the literature.
Author Response
Dear reviwer, Thank you very much for the comments on our manuscript. We take your feedback seriously and have done our best to revise the paper according to your comments. Please see the attachment. We sincerely hope to make our academic contributions with this continuously improved piece of work. Best Regards, Yiyun Hu

Reviewer 2 Report
Thanks for the manuscript. A few recommendations:
- The readability could be enhanced if the current long introduction could have a sub-section on context/literature.
- It is unclear what the significance of this study is (e.g., how do the factors impacting the sustainable development of professional learning communities beneficial for K12 education?)
- The limitation of the study as well as the possible future study are absent.
- It is not obvious how section 2 (analytical framework) is linked to research methodology (e.g., how is the framework being adopted in data analysis?)
- The discussion on results in related to literature could be strengthen (e.g., to what extent the results inform the existing literature findings?).
Author Response
Dear Reviewer, Thank you very much for the comments on our manuscript. We take your feedback seriously and have done our best to revise the paper according to your comments. Please see the attachment. We sincerely hope to make our academic contributions with this improved piece of work. Best Regards, Yiyun Hu

Round 2
Reviewer 1 Report
Thank you so much for your revision.
Author Response
Thank you too, for your kind effort in reviewing my work!
Reviewer 2 Report
Thanks for the revised manuscript by addressing all the feedback. Please send the manuscript to go through professional proof reading/editing service in order to enhance the readability.
Author Response
Thank you for your comments, the manuscript has now been through professional English editing, and we have proofread it carefully.